# Analysis of Volcanic Thermohaline Fluctuations of Tagoro Submarine Volcano (El Hierro Island, Canary Islands, Spain)

**Anna Olivé Abelló** [1,2], **Beatriz Vinha** [2], **Francisco Machín** [3], **Francesco Zerbetto** [4], **Evangelos Bakalis** [4,*] and **Eugenio Fraile-Nuez** [2,*]

1. Departament d'Oceanografia Física i Tecnològica, Institut de Ciències del Mar, CSIC, 08003 Barcelona, Spain; aolive@icm.csic.es
2. Centro Oceanográfico de Canarias, Instituto Español de Oceanografía (IEO), Consejo Superior de Investigaciones Científicas (CSIC), 38180 Santa Cruz de Tenerife, Spain; beatrizvinha95@gmail.com
3. Departamento de Física, Universidad de Las Palmas de Gran Canaria, 35017 Las Palmas de Gran Canaria, Spain; francisco.machin@ulpgc.es
4. Dipartimento di Chimica "G. Ciamician", Università di Bologna, 40126 Bologna, Italy; francesco.zerbetto@unibo.it
* Correspondence: evangelos.bakalis2@unibo.it (E.B.); eugenio.fraile@ieo.es (E.F.-N.)

**Abstract:** Temperature and conductivity fluctuations caused by the hydrothermal emissions released during the degasification stage of the Tagoro submarine volcano (Canary Islands, Spain) have been analysed as a robust proxy for characterising and forecasting the activity of the system. A total of 21 conductivity-temperature-depth time series were gathered on a regular high-resolution grid over the main crater of Tagoro volcano. Temperature and conductivity time series, as manifestations of stochastic events, were investigated in terms of variance and analysed by the Generalised Moments Method (GMM). GMM provides the statistical moments, the structure functions of a process whose shape is an indicator of the underlying stochastic mechanisms and the state of activity of the submarine volcano. Our findings confirm an active hydrothermal process in the submarine volcano with a sub-normal behaviour resulting from anti-persistent fluctuations in time. Its hydrothermal emissions are classified as multifractal processes whose structure functions present a crossover between two time scales. In the shorter time scale, findings point to the multiplicative action of two random processes, hydrothermal vents, which carries those fluctuations driving the circulation over the crater, and the overlying aquatic environment. Given that both temperature and conductivity fluctuations are nonstationary, Tagoro submarine volcano can be characterised as an open system exchanging energy to its surroundings.

**Keywords:** Tagoro submarine volcano; time series; volcanic activity; generalised moments method; stochastic processes

## 1. Introduction

Hydrothermal vents, found in submarine volcanic islands, release important emissions of hot fluids that rise in a buoyant turbulent plume over the source [1]. Those hydrothermal emissions range from intense high-temperature plumes to diffuse low temperature diluted discharges emanating directly from the seafloor [2]. The fluids mix with the surrounding seawater and rise to a level where their density matches the water outside the plume being able to alter the temperature and salinity fields specific to the vertical seawater structure [3]. These vertical motions generate a horizontal flow, resulting in an anticyclonic vortex of sufficient magnitude to trap water, minerals, tracers, and organisms in a local recirculation [4]. As a result, temperature and conductivity fields may fluctuate, influencing the local circulation. At the same time, fluctuations are the outcome of random perturbations acting on a field, noise sources caused by the short-term circulation over the volcano [5]. Decoding fluctuations with stochastic analysis tools can shed light and benefit understanding the complexity of hydrothermal emissions [6–8].

In the Mediterranean, a newly developed mathematical technique was applied to the time series of temperature and conductivity fluctuations, recorded by remotely operated vehicles (ROV), to understand the nature of the stochastic processes that drive the activity of volcanoes [6,7]. These studies suggest that temperature fluctuations can be used as an indicator to determine whether the volcano operates as an open or closed system, i.e., if it exchanges energy with the surroundings or not. Open stands for nonstationary temperature time series and closed for stationary ones. On the other hand, conductivity, the ability of a water sample to conduct an electrical current, fluctuations can provide information about its current state of activity. During unrest (active) degassing periods, conductivity and temperature follow a universal multifractal behaviour, i.e., a multiplicative action of random processes, with continuous energy dissipation. Otherwise, conductivity behaves as a universal multifractal but with a stationary pattern [6,7].

El Hierro is the youngest and southwesternmost island of the Canary Archipelago characterised by a three-armed rift system and originated by hotspot volcanism [9,10] (Figure 1). On 10 October 2011, a submarine volcano eruption took place 1.8 km south of the coast of El Hierro Island, forming the shallow submarine volcano Tagoro, located at 27°37.1160′ N, 017°59.4660′ W [11]. When the eruption finished in March 2012, hydrothermal manifestations were characterised by the release of hot fluids, gases, inorganic nutrients and metals as a result of the new degasification phase of the post-eruptive stage of the Tagoro submarine volcano [12]. This degasification process is exclusively associated with hydrothermalism linked with shallow submarine volcanoes, resulting in water properties changes [13,14].

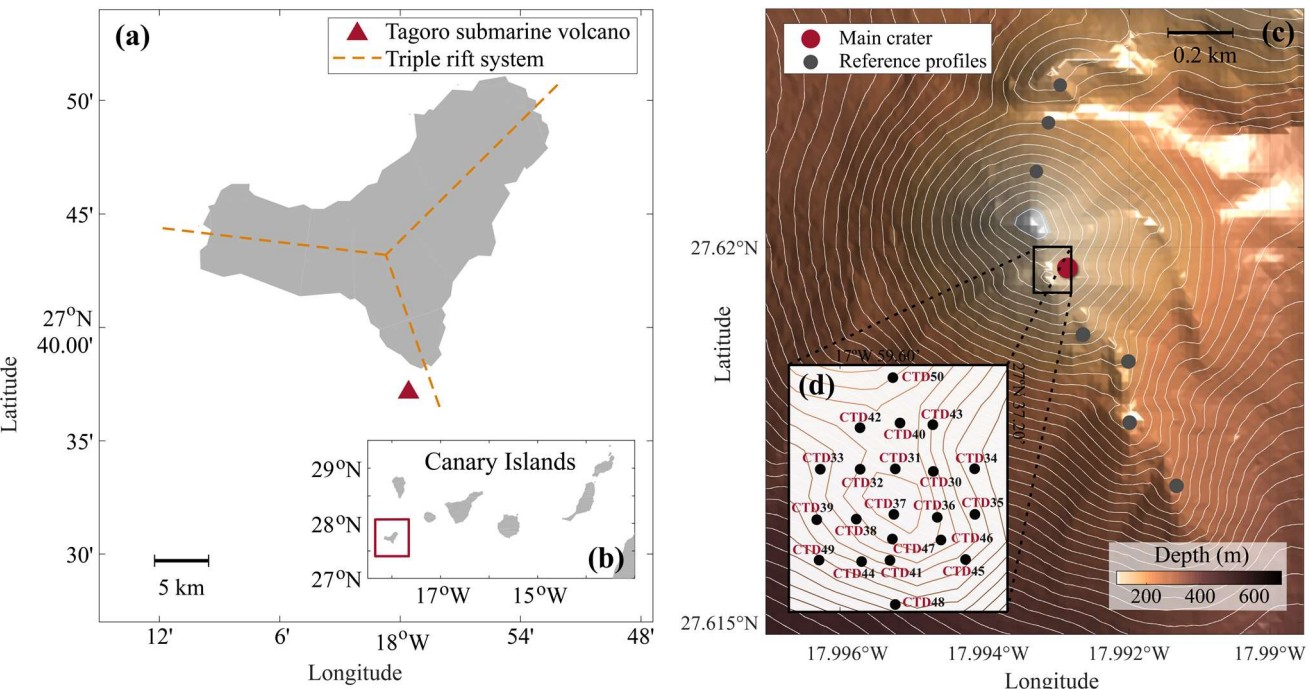

**Figure 1.** (**a**) Map of El Hierro Island with the triple rift system marked (dashed line) and the location of Tagoro submarine volcano (red triangle). (**b**) Location of El Hierro Island (red square box) in the Canary Islands Archipelago. (**c**) High-resolution bathymetric map of the Tagoro submarine volcano with the location of the main crater of 15 m in diameter (red dot), the reference profiles (grey dots) and (**d**) the distribution of the 21 CTD time series recorded along the crest of the seamount.

Since 2011, the Tagoro submarine volcano has been deeply monitored by more than 28 different multidisciplinary cruises in order to understand submarine volcanic processes better and how the volcano evolves in terms of different physical-chemical seawater parameters and the relationship of these with the surrounding marine ecosystem [8,11,12,15–27].

However, there is a big gap in knowledge in terms of understanding how fluctuations in temperature and conductivity can provide information regarding the hydrothermal system and its state of activity.

The main goal of this manuscript is, using stochastic calculus tools, to assess and classify the state of activity (rest/unrest) and characterise the nature of stochastic mechanisms pertaining to around and over the main crater of the Tagoro submarine volcano in its degassing stage. This objective will be evaluated by applying the mathematical methodology proposed in [6] and using the temperature and conductivity measurements collected by a CTD sensor (Conductivity-Temperature-Depth) mounted on a rosette over the Tagoro submarine volcano. In addition, this will be the first time that a cost-effective oceanographical instrument (rosette) is implemented to register statically hydrothermal emissions in the Atlantic Ocean instead of the ROV, widely used in the Mediterranean Sea [6,26,28]. Although ROVs are a valuable tool for collecting in situ samples and recording HD images, compared with the rosette sampler, with a significantly higher measurement frequency, they are much more limited.

## 2. Materials and Methods

In October 2016, the multidisciplinary cruise VULCANO-II-1016 on board of R/V *Ángeles Alvariño* was carried out. One of the survey objectives consisted of deploying up to 21 oceanographic stations with a CTD sensor mounted on the rosette sampler in a regular 10.5-m high-resolution grid above the main crater, main crest and surroundings (Figure 1d).

Each station records a unique time series of conductivity and temperature fluctuations, referred to here with the $E_c$ and $\theta$ parameters, between a time length from 9 to 13 min, with the exceptions of CTDs 41, 43 and 47, as a result of discarding some periods of fluctuations because the rosette was not static one meter above the seafloor (Table 1). Thus, each CTD time series was sampled with a frequency of 24 Hz or 0.0417 s distance between two consecutive measurements maintaining the rosette sampler, through the altimeter sensor, at approximately 1 m above the seabed. At the same time, the ship remained stationary on the surface through its Dynamic Positioning System. Data were acquired using an SBE 911-plus CTD equipped with dual temperature and conductivity sensors, calibrated at the SeaBird laboratory before and after the cruise, with accuracies of 0.001 °C and 0.0003 S/m, respectively.

**Table 1.** The location, duration and date of the 21 CTD time series recorded during the multidisciplinary cruise VULCANO-II-1016. The initial and final times correspond to the entire time in which the oceanographic rosette sampler remained in the water. The duration at the seabed (in minutes and seconds) corresponds to the length of the CTD time series fluctuations without the descending and ascending track. All the minutes until the oceanographic rosette reach the volcano edifice and remain steady one meter above the seafloor were discarded, resulting in significatively shorter time series for CTDs 41, 43 and 47.

| N° CTD | Latitude | Longitude | Initial Time | Final Time | Duration |
|---|---|---|---|---|---|
| **CTD30** | 27°37.1788′ N | 17°59.5831′ W | 28 October 2016 12:59 | 28 October 2016 13:17 | 11 min. 21 s |
| **CTD31** | 27°37.1793′ N | 17°59.5893′ W | 28 October 2016 13:38 | 28 October 2016 13:56 | 12 min. 20 s |
| **CTD32** | 27°37.1793′ N | 17°59.5944′ W | 28 October 2016 14:36 | 28 October 2016 14:54 | 9 min. 37 s |
| **CTD33** | 27°37.1788′ N | 17°59.6008′ W | 28 October 2016 15:14 | 28 October 2016 15:32 | 10 min. 44 s |
| **CTD34** | 27°37.1795′ N | 17°59.5775′ W | 28 October 2016 15:57 | 28 October 2016 16:15 | 11 min. 40 s |
| **CTD35** | 27°37.1725′ N | 17°59.5773′ W | 28 October 2016 16:36 | 28 October 2016 16:55 | 11 min. 49 s |
| **CTD36** | 27°37.1732′ N | 17°59.5836′ W | 28 October 2016 17:18 | 28 October 2016 17:37 | 12 min. 32 s |
| **CTD37** | 27°37.1729′ N | 17°59.5895′ W | 28 October 2016 17:57 | 28 October 2016 18:16 | 12 min. 17 s |
| **CTD38** | 27°37.1729′ N | 17°59.5257′ W | 28 October 2016 18:28 | 28 October 2016 18:47 | 13 min. 19 s |
| **CTD39** | 27°37.1731′ N | 17°59.6008′ W | 28 October 2016 19:42 | 28 October 2016 19:59 | 11 min. 25 s |

**Table 1.** *Cont.*

| N° CTD | Latitude | Longitude | Initial Time | Final Time | Duration |
|--------|----------|-----------|--------------|------------|----------|
| **CTD40** | 27°37.1849′ N | 17°59.5878′ W | 28 October 2016 20:16 | 28 October 2016 20:34 | 12 min. 1 s |
| **CTD41** | 27°37.1670′ N | 17°59.5904′ W | 28 October 2016 20:55 | 28 October 2016 21:13 | 5 min. 0 s |
| **CTD42** | 27°37.1848′ N | 17°59.5949′ W | 29 October 2016 07:10 | 29 October 2016 07:28 | 11 min. 35 s |
| **CTD43** | 27°37.1849′ N | 17°59.5936′ W | 29 October 2016 07:56 | 29 October 2016 08:13 | 6 min. 6 s |
| **CTD44** | 27°37.1674′ N | 17°59.5948′ W | 29 October 2016 08:42 | 29 October 2016 08:59 | 11 min. 28 s |
| **CTD45** | 27°37.1680′ N | 17°59.5792′ W | 29 October 2016 09:33 | 29 October 2016 09:50 | 10 min. 45 s |
| **CTD46** | 27°37.1704′ N | 17°59.5822′ W | 29 October 2016 10:51 | 29 October 2016 11:11 | 11 min. 25 s |
| **CTD47** | 27°37.1701′ N | 17°59.5895′ W | 29 October 2016 11:33 | 29 October 2016 11:52 | 6 min. 38 s |
| **CTD48** | 27°37.1616′ N | 17°59.5887′ W | 29 October 2016 12:23 | 29 October 2016 12:40 | 11 min. 28 s |
| **CTD49** | 27°37.1674′ N | 17°59.6005′ W | 29 October 2016 13:01 | 29 October 2016 13:18 | 9 min. 53 s |
| **CTD50** | 27°37.1909′ N | 17°59.5891′ W | 29 October 2016 13:48 | 29 October 2016 14:06 | 10 min. 46 s |

The time series of temperature ($\theta$) and conductivity ($E_c$) recorded at each location inside the grid are subject to fluctuations and have been analysed as the result of stochastic processes. In this sense, an initial evaluation of the dynamics of the hydrothermal emissions fluctuations driven by the activity of the volcano, $x(t)$, can be made by examining the time dependence of its variance, $W(t) = \langle x^2(t) \rangle - \langle x(t) \rangle^2$. If the process, $x(t)$, corresponds to uncorrelated random events (normal distribution), then the variance of $x(t)$ grows linearly in time (Gaussian behaviour). Any departure from linearity, i.e., anomalous behaviour, might indicate the existence of anti-correlated/correlated events and/or systems where the environment fluctuates at similar time scales as the random variable $x(t)$ [29]. In the anomalous regime, the behaviour is sub-normal/super-normal for growth slower/faster than linear, and the variance reads [30]:

$$W(t) = \frac{K_\gamma t^\gamma}{\Gamma(1+\gamma)} \qquad (1)$$

where $K_\gamma$ is a generalised coefficient expressed in proper units, e.g., for temperature the units are °C² s$^{-\gamma}$, and $\Gamma$ () is the Gamma function. According to the exponent $\gamma$, the stochastic processes can be classified as sub-normal (anti-persistent process) for values from $0 < \gamma < 1$, Brownian or normal for $\gamma = 1$, super-normal (persistent process) for $1 < \gamma < 2$, ballistic for $\gamma = 2$, and stationary for $\gamma = 0$. It is worth mentioning that Equation (1) does not apply to instantaneous changes of enormous amplitude (Lévy flights) where the first moment diverges. For discrete time series, Equation (1) is expressed as time average and reads:

$$W(\Delta, T) = \frac{1}{M} \sum_{i=1}^{M} \frac{1}{T-\Delta} \sum_{n=1}^{T-\Delta} (x_i(n+\Delta) - x_i(n))^2 - \left( \frac{1}{M} \sum_{i=1}^{M} \frac{1}{T-\Delta} \sum_{n=1}^{T-\Delta} (x_i(n+\Delta) - x_i(n)) \right)^2 \quad (2)$$

where $T = N\tau$ is the total time, $N$ is the total number of measurements, and $\tau = 0.0417$ s is the reciprocal of the maximum sampling rate. $M$ represents the number of time series monitored under the same conditions, the first summation being omitted when $M = 1$. The parameter $\Delta$ is the lag time and plays the role of the actual time in the analysis; it takes values in the range $\tau \leq \Delta \leq T/10$. Temperature or conductivity time series are represented by $x_i(n)$ with $n = 1, 2, 3, \ldots, N$ and $i = \theta$ or $E_c$.

Equation (2) provides an adequate description of when the process is normal, $\gamma = 1$. Any departure from normality requires the estimation of more moments, including fractional ones. We apply the Generalised Moments Methods (GMM) following these required steps:

- Construction of $N/10$ new time series each of length $T - \Delta$. The new time series, $y_n(\Delta)$, contains the absolute change between two values of the original series that are set apart by $\Delta$, so that $y_n(\Delta) = |x(n\tau + \Delta) - x(n\tau)|$ for $n = 1, 2, \ldots, ((T - \Delta)/\tau)$ and for $\Delta = m\tau$ with $m = 1, 2, ..., N/10$ where $\tau$ and $N\tau/10$ define the minimum and maximum time lags.
- Estimation of the statistical moments, order of $q$, for each one of the new time series, $y_n(\Delta)$, is carried out according to:

$$\rho(q, \Delta) = \frac{1}{T - \Delta} \sum_{n=1}^{T-\Delta} (y_n(\Delta))^q \tag{3}$$

where only positive moments, $q$, are taken into account. Moments between $0 < q \leq 2$ are responsible for the core of the probability density function (pdf), while moments $q > 2$ contribute to the tails of the pdf (Bakalis et al., 2017 and the references therein).
- The moments will scale according to:

$$\rho_m(q, \Delta) \approx \Delta^{z(q)} \tag{4}$$

where $z(q)$ is the structure function and its shape gives information about the stochastic mechanisms that govern the process. When the structure function fits a linear form, that is, $z(q) = Hq$, then there is a direct relation between the scaling exponent $\gamma$ of Equation (1) and the Hurst exponent, $H$, $\gamma = 2H$, being the process characterised as monofractal. The parameter $H$ is the mean fluctuating scaling exponent. For $0 < H < 0.5$ the process is anti-persistent (sub-normal), normal for $H = 0.5$, and super-normal or persistent for $0.5 < H < 1$. If $z(q)$ presents any convex shape, then the process is multifractal. Among multifractals, universal multifractals are ubiquitous, and their structure function reads [31]:

$$z(q) = Hq - \frac{C}{a - 1}(q^a - q) \tag{5}$$

where the Lévy index, $a$, $0 \leq a \leq 2$, indicates the class to which the probability distribution belongs and shows how fast the inhomogeneity rises. Inhomogeneity refers to the degree of mixing of various stochastic mechanisms shaping the final process. For $a = 0$, $z(q) = Hq$, the process is monofractal, and the field homogeneous. For $a = 1$, $z(q) = Hq - Cq\log(q)$, the process draws steps from a Cauchy–Lorentz distribution. For $a = 2$, $z(q) = Hq - C(q^2 - q)$, the process is the multiplication of two random processes and is described by a log-normal or Kolmogorov distribution. For $1 < a < 2$, the multifractal character is the multiplicative result of more than two random processes. Finally, $C$ is the intermittency parameter that measures the mean inhomogeneity of the field and always takes positive values.

## 3. Results

Implementing a high-resolution grid of temporal series around and over the main crater reveals temperature and conductivity fluctuations in the whole domain, suggesting that the fields are disturbed by emissions from the volcano and may be far from a thermodynamic equilibrium. Using non-affected vertical profiles of $\theta$ (°C) and $E_c$ (mS/cm) measured simultaneously on the submarine volcano surroundings (see Figure 1c), we obtained the average temperature and conductivity reference profiles and their standard deviation down to 127 m (shown in Figure 2). The high variations of both properties during the whole time series over the seabed, concerning the reference profile's standard deviation, confirm the significance of the fluctuations due to the continuous release of hydrothermal emissions. These fluctuations as a function of time and their correlations for the randomly selected CTD30, one of the 21 time series collected with a time interval of 11 min and 21 s, are illustrated in detail in Figure 3.

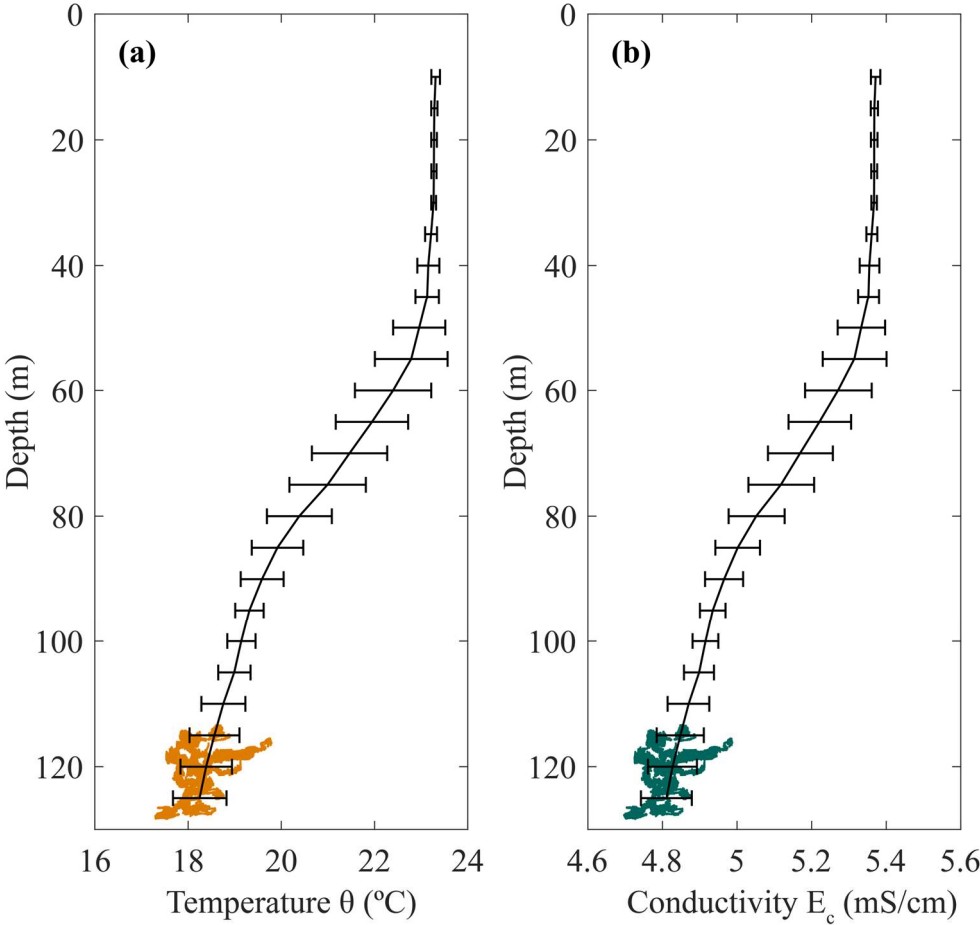

**Figure 2.** Mean vertical profile of (**a**) temperature, θ, and (**b**) conductivity, E$_c$, with their standard deviation used as a reference from outlying stations not affected by the submarine volcano emissions. The time series fluctuations measured around and over the main crater, in orange for temperature and dark green for conductivity, are located close to 127 m depth.

The variances of temperature and conductivity have been obtained as a function of lag time for all CTD time series. Figure 4 shows the variance of both temperature (Figure 4a) and conductivity (Figure 4b) of the CTD30 sampling point. Both properties present a quite similar scaling with a crossover point at $\Delta \approx 3.5$ s, defined as the time where the scaling exponent changes. The crossover point divides the time series into two time scales or regimes: the shorter time scale corresponds to the first regime and small lag times, and the longer time scale considers all the lag times and therefore, we will refer to it as the whole regime. We fitted each curve with Equation (1), $W(\Delta) = b\Delta^{\gamma}$, obtaining the early (small lag times) and the long-time (all lag times) behaviour for θ and E$_c$. For small lag times, $0.042 < \Delta \leq 3.5$ s, temperature and conductivity pose similar weak sub-normal behaviour, with scaling exponents 0.84 and 0.85, respectively. For the long-time behaviour, the whole lag times range has been considered. Again, both θ and E$_c$ scale similarly, with values of exponents $\gamma = 0.75$ and $\gamma = 0.77$, respectively. For both small and large times, temperature and conductivity follow sub-normal behaviour, anti-persistent variations, which become stronger for longer times and lesser when scaling exponents. The absence of stationary behaviour ($\gamma = 0$) in temperature confirms the presence of a mechanism that causes the alterations (see also below), suggesting the existence of a remarkable hydrothermal vent field, consistent with what has been described so far in the area [8,21].

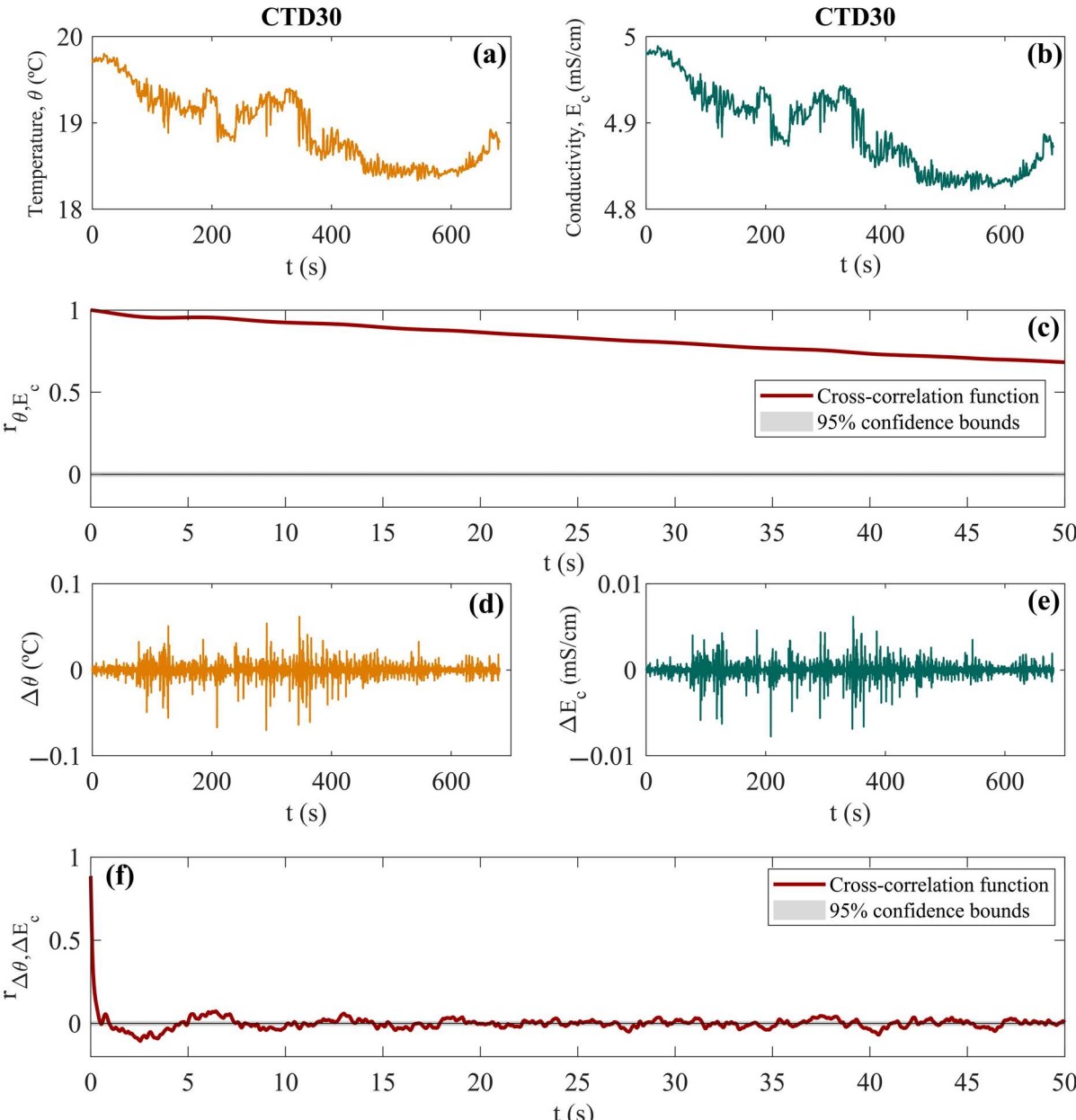

**Figure 3.** (**a**) Temperature, θ, and (**b**) conductivity, E$_c$, time series of CTD30 collected as part of the high-resolution grid. The similarity between both variable fluctuation patterns as well as the occasional distinct peaks is remarkable, and (**c**) their normalised cross-correlation coefficient r$_{θ,Ec}$ showing a strong correlation for short times (t < 3.5 s), which remains large even for long times. (**d**) The differentiation with respect to time of the temperature time series, Δθ, (**e**) the differentiation with respect to time of the conductivity time series, ΔE$_c$, and (**f**) the normalised cross-correlation coefficient r$_{Δθ,ΔEc}$ of the differentiation with respect to time of temperature and conductivity time series going rapidly to zero, at the crossing point 0.6 s. In grey is indicated the 95% confidence bounds, where the time series are uncorrelated.

The rest of the CTD time-series exhibit crossover points whose positional values range from 1.8 to 5.04 s. Their variances for the whole regime are consistent with the CTD30 sub-normal behaviour, always with scaling exponent in the range $0 < γ < 1$ (Table 2). In the short time scale (first regime), although most of the sampling points show a similar pattern in terms of variance and can be defined as sub-normal, the temperature variances for CTDs 37, 38, and 50 and conductivity variances for CTDs 32, 38, 40, 48, and 50 suggest in those sampled locations processes that can be characterised as Brownian ($γ ≈ 1$), since $0.95 ≤ γ$.

The latter highlights the necessity of a closer investigation of the analysis of the recorded data which extends beyond the standard variance.

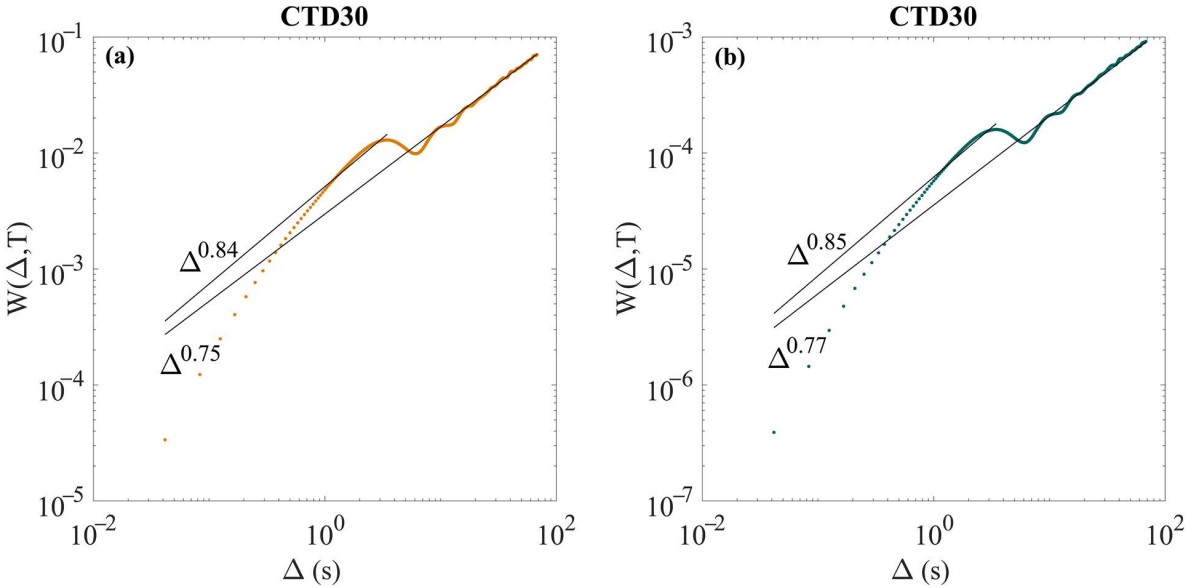

**Figure 4.** Logarithmic plots of the variance for (**a**) temperature, θ, (orange dots) and (**b**) conductivity, $E_c$, (dark green dots) with respect to the lag time for the CTD30 time series. The variance for both temperature and conductivity has been fitted by Equation (1). In each figure, the short black solid line represents the best fit for short times (first regime), and the long one the best fit for long times (whole regime). Notice that Δ (s) refers to the unit of time in seconds.

**Table 2.** $γ$, $α$, $H$, and $C$ parameters of the temperature and conductivity for both first and whole regimes (short and long times) and for each one of the recorded CTD time series. Notice that for short times the alpha index is 2.

| | Temperature θ | | | | | | | | Conductivity $E_c$ | | | | | | | |
| | First Regime | | | | Whole Regime | | | | First Regime | | | | Whole Regime | | | |
| | $γ$ | $α$ | $H$ | $C$ | $γ$ | $α$ | $H$ | $C$ | $γ$ | $α$ | $H$ | $C$ | $γ$ | $α$ | $H$ | $C$ |
|---|---|---|---|---|---|---|---|---|---|---|---|---|---|---|---|---|
| CTD30 | 0.84 | 2 | 0.662 | 0.055 | 0.75 | 1.610 | 0.424 | 0.059 | 0.85 | 2 | 0.669 | 0.054 | 0.77 | 1.578 | 0.434 | 0.060 |
| CTD31 | 0.80 | 2 | 0.620 | 0.041 | 0.50 | 1.045 | 0.352 | 0.099 | 0.81 | 2 | 0.628 | 0.045 | 0.51 | 1.078 | 0.360 | 0.100 |
| CTD32 | 0.91 | 2 | 0.738 | 0.046 | 0.55 | 1.159 | 0.316 | 0.085 | 0.99 | 2 | 0.730 | 0.052 | 0.61 | 1.428 | 0.330 | 0.068 |
| CTD33 | 0.78 | 2 | 0.598 | 0.032 | 0.85 | 1.79 | 0.403 | 0.053 | 0.90 | 2 | 0.620 | 0.033 | 0.87 | 1.793 | 0.391 | 0.052 |
| CTD34 | 0.78 | 2 | 0.557 | 0.046 | 0.44 | 1.076 | 0.316 | 0.065 | 0.90 | 2 | 0.571 | 0.048 | 0.46 | 1.104 | 0.324 | 0.063 |
| CTD35 | 0.74 | 2 | 0.564 | 0.025 | 0.42 | 1.136 | 0.382 | 0.124 | 0.75 | 2 | 0.578 | 0.028 | 0.42 | 1.188 | 0.384 | 0.117 |
| CTD36 | 0.80 | 2 | 0.601 | 0.028 | 0.57 | 1.115 | 0.446 | 0.180 | 0.75 | 2 | 0.593 | 0.019 | 0.62 | 1.208 | 0.456 | 0.168 |
| CTD37 | 0.96 | 2 | 0.619 | 0.056 | 0.74 | 1.622 | 0.405 | 0.048 | 0.82 | 2 | 0.631 | 0.061 | 0.79 | 1.633 | 0.422 | 0.047 |
| CTD38 | 0.96 | 2 | 0.622 | 0.047 | 0.55 | 1.190 | 0.390 | 0.067 | 0.96 | 2 | 0.644 | 0.055 | 0.66 | 1.171 | 0.406 | 0.065 |
| CTD39 | 0.73 | 2 | 0.568 | 0.021 | 0.80 | 1.199 | 0.469 | 0.109 | 0.86 | 2 | 0.588 | 0.024 | 0.83 | 1.244 | 0.483 | 0.110 |
| CTD40 | 0.83 | 2 | 0.564 | 0.004 | 0.39 | 1.395 | 0.295 | 0.105 | 1.01 | 2 | 0.589 | 0.018 | 0.42 | 1.480 | 0.306 | 0.105 |
| CTD41 | 0.85 | 2 | 0.604 | 0.058 | 0.26 | 1.275 | 0.307 | 0.102 | 0.84 | 2 | 0.627 | 0.058 | 0.29 | 1.314 | 0.324 | 0.100 |
| CTD42 | 0.93 | 2 | 0.614 | 0.021 | 0.59 | 1.796 | 0.430 | 0.047 | 0.94 | 2 | 0.631 | 0.017 | 0.71 | 1.821 | 0.446 | 0.055 |
| CTD43 | 0.85 | 2 | 0.623 | 0.05 | 0.62 | 1.367 | 0.426 | 0.126 | 0.84 | 2 | 0.642 | 0.056 | 0.63 | 1.391 | 0.438 | 0.123 |
| CTD44 | 0.77 | 2 | 0.598 | 0.064 | 0.61 | 1.210 | 0.356 | 0.051 | 0.85 | 2 | 0.623 | 0.075 | 0.53 | 1.223 | 0.368 | 0.055 |
| CTD45 | 0.88 | 2 | 0.620 | 0.013 | 0.88 | 1.902 | 0.416 | 0.051 | 0.90 | 2 | 0.646 | 0.022 | 0.90 | 1.847 | 0.430 | 0.056 |
| CTD46 | 0.77 | 2 | 0.540 | 0.039 | 0.52 | 2.000 | 0.345 | 0.017 | 0.82 | 2 | 0.573 | 0.048 | 0.77 | 2.000 | 0.358 | 0.018 |
| CTD47 | 0.73 | 2 | 0.559 | 0.032 | 0.57 | 1.547 | 0.356 | 0.055 | 0.77 | 2 | 0.599 | 0.039 | 0.59 | 1.540 | 0.377 | 0.061 |
| CTD48 | 0.88 | 2 | 0.619 | 0.034 | 0.43 | 1.431 | 0.393 | 0.162 | 0.99 | 2 | 0.652 | 0.045 | 0.21 | 1.445 | 0.405 | 0.163 |
| CTD49 | 0.77 | 2 | 0.584 | 0.036 | 0.59 | 1.000 | 0.536 | 0.130 | 0.79 | 2 | 0.600 | 0.043 | 0.62 | 1.000 | 0.545 | 0.134 |
| CTD50 | 0.97 | 2 | 0.662 | 0.037 | 0.15 | 0.941 | 0.237 | 0.122 | 1.07 | 2 | 0.695 | 0.044 | 0.14 | 0.928 | 0.283 | 0.131 |

Generalised moments for both temperature and conductivity measurements up to the fourth order were determined by using Equation (3) and the moments for the CTD30 time series are illustrated in Figure 5. A vertical solid line corresponds to the crossover point dividing the two time scales, and its choice is based on the turnover point of the variance. For both θ and $E_c$, each moment shows dependence on the time lags.

Each moment has been fitted by Equation (4), $\rho(q, \Delta) \approx \Delta^{z(q)}$, and the obtained exponent is equal to the value of the structure function for this specific moment. In this way, the values of $z(q)$ for the moments defined in the range $0.25 \leq q \leq 4$ will be fitted later by Equation (5) for the estimate of the structure function. For the CTD30, the structure function for each property is represented in Figure 6. The value $z(q)$ has a convex shape as a function of the order of the moment, $q$, for both short and long times. This convexity indicates the multifractality of temperature and conductivity time series. For the first regime (short times), the Lévy stability index, $a$, of the structure function is equal to 2 and points to the existence of a log-normal or Kolmogorov distribution, which describes the multiplicative effect of two random processes. In the simplest scenario, the first process is likely the volcano hydrothermal vents that would drive the second one, a short-term circulation over the crater that contributes to the rise of the temperature and conductivity variables. For longer times, the process becomes even more complicated. The stability index takes values $a = 1.61$ and $a = 1.58$ for temperature and conductivity, respectively, highlighting an increase in complexity as the observation time increases. And yet, this increase in complexity may reflect the existence of more than the previously mentioned two random processes that shape the overall multifractal character. The same is true for all CTD time series analysed in the vicinity of the crater because the structure function for short times always presents a log-normal distribution with Lévy stability index equal to 2 and for long times corresponds to a universal multifractal, Equation (5), whose Lévy stability index takes value in the range $1.04 \leq a < 1.90$, see Table 2.

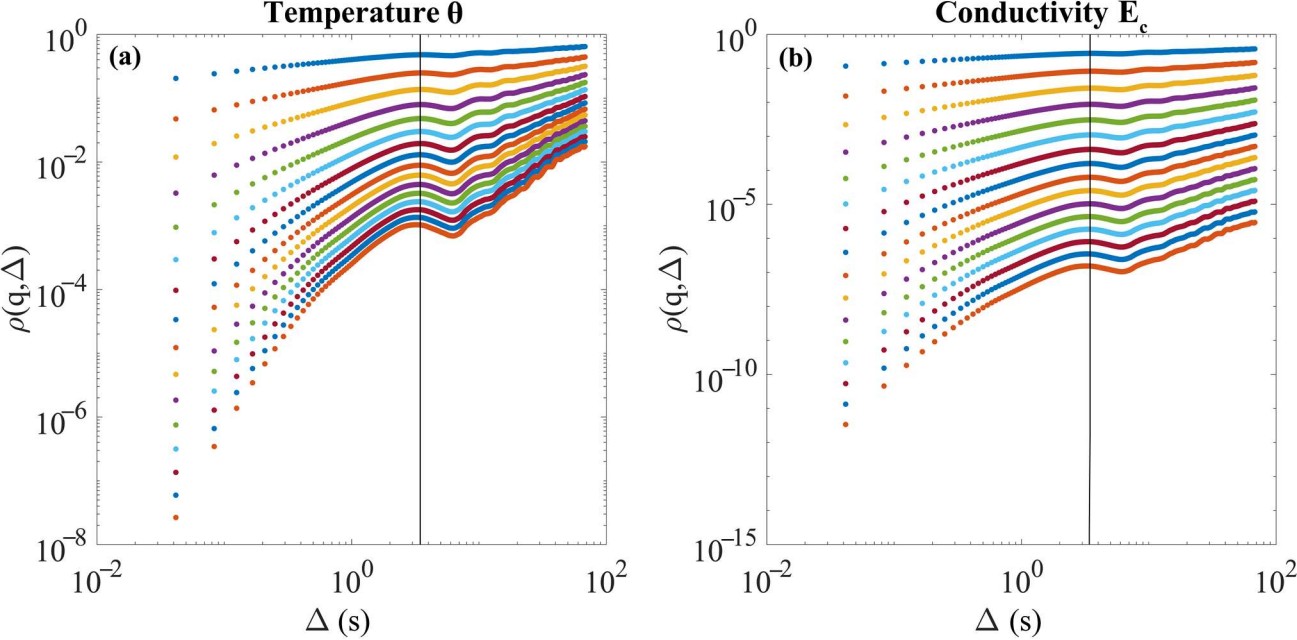

**Figure 5.** Generalised moments of CTD30 for (**a**) temperature, θ, and (**b**) conductivity, $E_c$, as function of the lag time, Δ, for different moment orders, $q$, ranging from 0.25 to 4 with steps of 0.25. The line represents the turning point where the convex shape of the short lag times ends. Notice that Δ (s) refers to the unit of time in seconds.

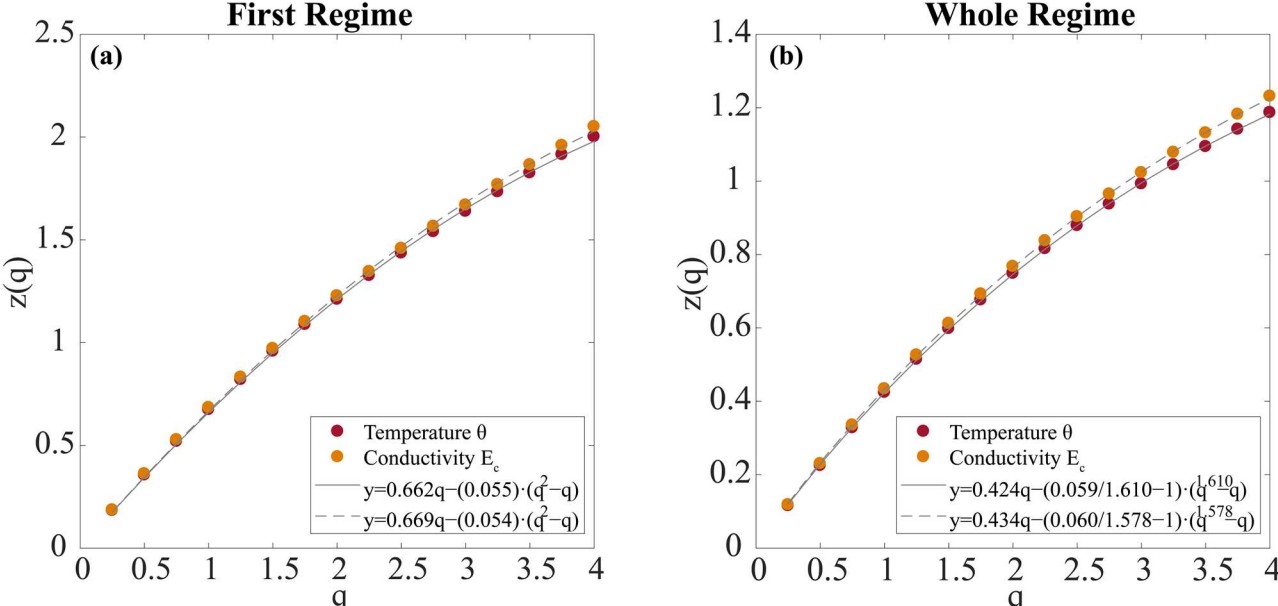

**Figure 6.** The structure function $z(q)$ versus the order of the moment, $q$, for (**a**) the first regime (short times) and (**b**) the whole regime (long times) for temperature, $\theta$, and conductivity, $E_c$, time series and for the CTD30 data.

For short times (Figure 6), the *H* and *C* parameters of the structure function of CTD30 take values of 0.662 and 0.055 for $\theta$ and 0.669 and 0.054 for $E_c$, respectively. Considering the structure functions obtained for all the CTD time series for the first regime ($a = 2$), listed in Table 2, the *H* exponent is varied between 0.540 and 0.695, implying super-normal processes if the intermittency parameter, parameter *C*, were zero. However, the latter is not true, and the non-zero intermittent effects establish conductivity and temperature mean fields as non-conservative. In the long-time limit, the structure-function of CTD30 provides values of *H* and *C* of 0.424 and 0.059 for $\theta$ and 0.434 and 0.060, for $E_c$, respectively. For long times, the Hurst exponents are lower than 0.5, implying stronger sub-normal processes and an increasing complexity reflected in the value of the alpha-stable index, which is different now for temperature and conductivity. For short times, the structure functions of both temperature and conductivity have a high similarity indicating a strong correlation of one another. Indeed, the normalised cross-correlation coefficient of the two, $r_{\theta,E_c}$, is higher than 0.95 for short times and decreases as the time increases (Figure 3c). Strong correlation does not necessarily imply the existence of causality between temperature and conductivity since this is the result of a broader mechanism affecting both properties equally and simultaneously, as the hydrothermal field does. Actually, by differentiating, with respect to time, the time series of temperature, $\Delta\theta$, and conductivity, $\Delta E_c$, (Figure 3d,e), respectively, the normalised cross-correlation coefficient of the two, $r_{\Delta\theta, \Delta E_c}$, goes rapidly to zero (it is not exactly delta correlated) and the crossing of the abscissa is at 0.6 s. After that point and for a short window, before fluctuations start around zero (see Figure 3f), $\Delta\theta$ and $\Delta E_c$ are anticorrelated to one another. These findings support the argument that the hydrothermal field is the reason for the strong correlation between temperature and conductivity since it operates as a vehicle carrying material to a sampling point.

The findings show that for both short and long time scales as defined above, the volcano area operates as an open system releasing a continuous flux of energy into the seawater, establishing thus temperature and conductivity to be non-conservative mean fields as a result of local swirls and/or vortexes. Alternatively, a non-conservative conductivity mean field might also be due to the presence of a magnetic field generated by the ions discharged from the crust of the volcanic hydrothermal chimneys to the water [6].

To understand how the environment and seawater above Tagoro main crater are affected by those mechanisms governing the stochastic process and characterising the

system, the spatial distribution of CTD time series according to its stochastic behaviour in the first regime (short times) is shown in Figure 7. There is not a significant trend in stochastic processes behaviour concerning the main crater distance, which is located over and between the CTDs 30, 34, 35 and 36. Analysing together the scaling of the mean field of temperature and its level of complexity, we found that the entire area, and not only the area above the crater, presents a higher activity and non-conservative field acting as an open system far from equilibrium. Near the crater, a less non-conservative region is found, characterised by an *H* smaller than 0.57. Additionally, in the middle of the high-resolution grid, the conductivity field can be considered an almost conservative one, *C* ~ 0 (Figure 7). These findings likely reflect local structures which do not allow strong mixing with the overlying surrounding environment, or it might mirror a stronger discharge of the fluid near the crater that weakens the effect of the marine short-term circulation at the sampling point in the window of observation, or it can be a combination of the two. The overall result reduces the intermittency parameter. Located mainly in the grid edges, the CTDs 40, 48, and 50, whose short time behaviour in terms of variance points to Brownian motion, present a multifractal character as well and match with a lower intermittency parameter, *C*, shown in Table 2.

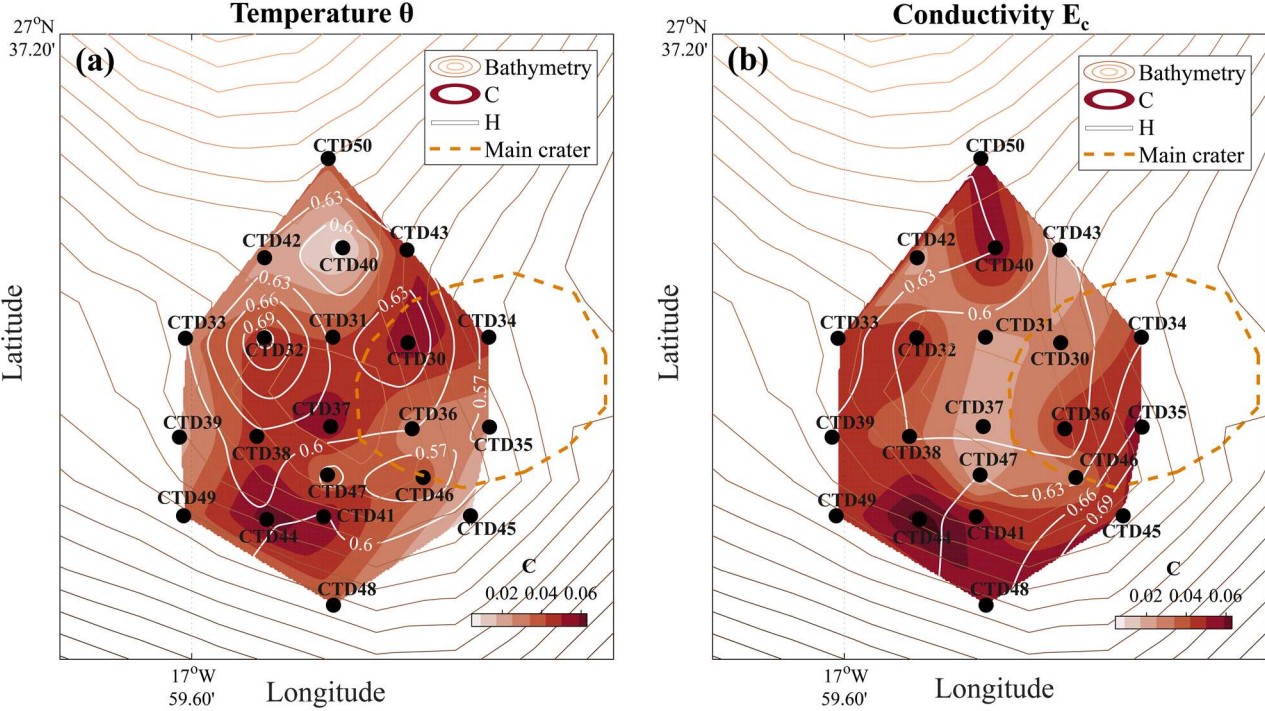

**Figure 7.** High-resolution grid distribution of (**a**) temperature, θ, and (**b**) conductivity, E$_c$, measurements above of the main crater. The black dots indicate the position where the CTD time series were registered over the seafloor. The filled contours represent the *C* parameter, while the white contours show the values of the Hurst parameter. The main crater (dashed orange line) is located over the CTDs 30, 34, 35, and 36.

In the extremes of the grid, the intermittency parameter of the conductivity field increases, highlighting higher complexity and displays the opposite behaviour for the temperature field. Possible contributions to the conductivity field at each sampling point can come directly from the volcano, and/or from the crust, and/or from diffusive ions from nearby points, and/or from the drifting of physical-chemical plumes, and/or from entrainment processes. If volcanic activity were the only source that affects the conductivity field, higher values of intermittency, *C*, would describe a non-continuous outgassing process reminiscent of short volcano breaths followed by longer ones, whereas low or even zero values would describe a continuous flow of material whose *H* value provides the risk of eruption (high risk for *H* ~1, ballistic behaviour, and low risk for *H* ~0.5, random behaviour).

The hypothesis of diffusivity as a source that affects the conductivity field is also discarded because it is not able to wash out local inhomogeneities (e.g., a typical diffusion coefficient of $10^{-9}$ m$^2$/s for ions moving in water during our observations results only in a displacement of ~1.3 mm) and to create a homogeneous one in such a reduced time of observation. Dynamic drifting is always present in open oceans, and its contributions might differ at different sampling points, thus mirroring the structure of the submarine volcano. The existence of drift is compatible with the variations of the temperature field. At sampling points where intermittency is almost zero for the conductivity field, its corresponding value for the temperature field is high indicating a strong mixing process. Certainly, there are points where intermittency takes high values for both fields mirroring strong mixing/competition of various random processes at these locations.

## 4. Discussion

The non-conservative mean fields are rather a consequence of the action of multiplicative noise, which can affect a physical system in many ways. To mention some of them, multiplicative noise can (i) stabilise a transient state [32], (ii) affect growth rates [33], and (iii) turn stationary distributions to non-Gaussian ones [34]. Furthermore, the strength of a multiplicative noise can modify the response of a system and unravel hidden periodicities, in the case that they exist, due to stochastic resonance [35]. Moreover, it is demonstrated that the effect of the buoyant seawater plume on the surrounding environment, together with the entrainment effect from the background waters and the general convective circulation over the hydrothermal sources, could potentially induce rapid changes in the temperature and conductivity mean fields [4]. This demonstrates that those physical-chemical anomalies or organisms remain near the source around the Tagoro system. Other physical oceanic processes, such as waves, tides, internal tides, etc., influence the variability of the temperature and conductivity mean fields at the Tagoro system on a larger time scale; however, a detailed study to understand such potential processes will be considered for futures studies.

The use of a high-resolution grid (Figure 7) allows us (i) to unveil the sub-normal behaviour of both temperature and conductivity fields in the entire area, and not only inside the main crater whose alterations are detected even several meters above the seafloor, (ii) to designate potential mechanisms for the functionality of the volcano, such as intermittency describing the outgassing ions and drift terms redistributing this material in the entire area, and (iii) to characterise the volcano as an active one, since multifractal properties of the conductivity field show that the volcano continuously provides material to the marine environment, and that it yet operates as an open system because of the non-conservative temperature field. The results confirm that the Tagoro submarine volcano is an active vent system significantly altering the properties of the seawater surrounding the main crater. Such physical-chemical changes have often been documented as changes in the temperature, salinity, pH, and ORP [8] in the pH and carbonate system [12,36], in the emission of reduced species and O$_2$ [11], in nutrient enrichment [27], and also in biological changes in terms of species composition [11,16,26], mainly related with bacterial and planktonic groups [20,22]. In agreement with the high fluctuations observed in the CTD profiles and the emerging gasses and particles documented by ROV videos, our results show that the system is far from both equilibrium and homogeneity.

The observed multifractality needs a closer look at the causes that produce it. A system driven by noise sources, multiplicative and/or additive, is usually described by a general class of stochastic differential equations (see Equation (1) of [37]). Alternatively, complex systems can be described by (i) fractional calculus and generalised fractional Langevin equations [38,39], (ii) linear Langevin equation subject to multiplicative and/or to additive noise [34], and (iii) stochastic population models [40–42]. A stochastic approach and description of the present system should take into account the findings of fluctuation analysis. We leave this task for forthcoming work. It is worth mentioning that the method (GMM) analytically described in the present work is not the only one that can be used for

fluctuations analysis. Methods such as detrended fluctuation analysis (DFA) [43], multi-fractal detrended fluctuation analysis (MFDFA) [44], diffusion entropy analysis (DEA) [45], or even the classical rescaled analysis (RA) [46] can be used if the corresponding time series is stationary.

For the first time, our study applied the methodology described by [6] to a shallow submarine volcano in the Atlantic Ocean. Previous studies had used time series data collected by a remotely operated vehicle (ROV) in Avyssos (Nisyros Island) [7], and in Kolumbo (Santorini Island) [6], a shallow submarine volcano, whose last eruption happened in 1650 [47]. The results of the study [6] showed that Kolumbo is a far-from-equilibrium system following sub-normal behaviour, similar to the results obtained in the present study for the Tagoro's hydrothermal system. For Kolumbo the crossover point, where the scaling exponent changes, is 5 s, similar to 3.5 s for the present study; however, a difference in the dynamics between the two (Kolumbo and Tagoro) might not be excluded due to closed/open sea conditions.

Overall, our results show that the GMM methodology can be applied to different systems and that the use of other oceanographical instruments for data collection is consistent with the effectiveness of the methodology. This can be useful since it allows choice of the instrument to be employed, although the ROV represent higher cost-effectiveness related to a rosette sampler since the utilisation of ROVs in oceanographical campaigns presents a notably higher cost.

The cost-effective methodology applied in this study may be exploited for forecasting future eruptions. By using this sampling methodology in time and using GMM for method analysis, we can characterise the level of activity (values of $H$ and $C$ for both temperature and conductivity) and monitor the evolution of such a physical system, predicting the natural hazard according to its potentiality. A combination of a high value of $H$ and a value of $C$ close to zero (almost homogeneous process, which means that the hydrothermal material dominates over the other processes) underlines an intense activity of the volcano, possibly according to a high hazard. On the other hand, for values of $H$ close to 0.5 and non-vanishing values of intermittency ($C$ parameter), the hazard is low since random environmental processes can affect the direct hydrothermal vent turning. The validity of the behaviour of $H$ and $C$ as prediction indexes should be tested in future experiments, which will monitor the dynamics of the El-Hierro submarine complex. These values of $C$ and $H$ are likely of importance for the existence of aquatic life in volcano surroundings, but extensive simulations in terms of fractional Langevin equations should be done to see how these parameters, especially $C$, affect the system. Such an analysis will deliver the memory kernel of the temperature and conductivity, examine possible repeated patterns, and help to study the future behaviour of Tagoro submarine volcano, see [6].

## 5. Conclusions

The Tagoro shallow submarine volcano, located south of El Hierro Island at the Canary Archipelago, has been deeply monitored in the last ten years since its origin. Its emissions, recorded with high sensitivity temperature and conductivity sensors mounted on an oceanographic rosette, exhibit important fluctuations in the surrounding of the main crater. The 21 $\theta$-$E_c$ time series have been analysed by the GMM method in a short window of observation, delivering a wealth of findings. GMM returns structure functions with convex shapes indicating that both temperature and conductivity fluctuations correspond to multiplicative processes. This behaviour indicates the direct competition between the overlying marine short-term circulation and the degassing material of the volcano leading to long-lived non-equilibrium states. The nonstationary nature of the temperature field points to an open system that releases energy continuously to the environment. The estimated $H$ and $C$ parameters showed that the risk of eruption is low at the corresponding time window. However, continuous monitoring of the volcanic activity is necessary to further compare and classify the future volcano activity. The latter can be done by using the

metrics (moments) analysed in the present work, which can be part of a machine learning approach aiming at forecasting the future activity of the Tagoro volcano.

**Author Contributions:** A.O.A., F.M. and E.F.-N.: data processing, initial mathematical analysis and graphical representations. E.B. and F.Z.: mathematical methodology verification. B.V, A.O.A., E.B., E.F.-N. and F.M.: writing–original draft preparation. B.V, A.O.A., F.M., E.B., F.Z. and E.F.-N.: writing–draft contribution. E.F.-N., F.M. and E.B.: supervision. E.F.-N.: project administration and funding acquisition. All authors have read and agreed to the published version of the manuscript.

**Funding:** This research was funded by FEDER and the Spanish Ministry of Economy and Competitiveness through the VULCANO-II project (CTM2014-51837-R) and for the Spanish Institute of Oceanography through the VULCANA-I (IEO-2015-2017) project, both projects led by the Spanish Institute of Oceanography (IEO), and by Fundación CajaCanarias-LaCaixa through project Cyclovent (2019SP18). Moreover, the Swiss company ORIS Höltsein 1904 funded the scholarship of one of the authors in the Spanish Institute of Oceanography facilities making possible the data preparation, data assessment and training.

**Data Availability Statement:** Data available under request.

**Acknowledgments:** We would like to thank the crew of the R/V *Ángeles Alvariño* and R/V *Ramón Margalef* for their assistance during the multiple oceanographic cruises.

**Conflicts of Interest:** The authors declare no conflict of interest. The funders had no role in the design of the study; in the collection, analyses, or interpretation of data; in the writing of the manuscript, or in the decision to publish the results.

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
