# Peer review of "Analysis of Volcanic Thermohaline Fluctuations of Tagoro Submarine Volcano (El Hierro Island, Canary Islands, Spain)"

_geosciences, doi:10.3390/geosciences11090374_

Round 1

Reviewer 1 Report

Dear Authors,

I have just completed the revision of your manuscript entitled “Analysis of volcanic thermohaline fluctuations of Tagoro submarine volcano (El Hierro Island, Canary Islands, Spain)”.

The article is well written, well balanced with a fluent English style. The mathematic/statistic part well support the results. As a first major remark ,  the paper lack of a real geologic/volcanologic and/or oceanographic section. I suggest you to include a brief description of the geologic setting.

Your study is intriguing and I am pretty  convinced that it would be worthy to be  published on Geosciences, anyway I have some major remarks:

1)The article is focused on a series of CTD casts distributed over the main crater region of the seamount. The acquisition was performed using classic rosette system. I would like to understand if the authors have considered  error of positioning and the fluctuation of the depth  before to use the data for statistical analysis. Dynamic positioning of the ship can not minimize heave movement which results in variation of depth position of the rosette. This variation would affect the data especially in case of Temperature.

2)I am wondering some additional information about the duration of single CTD cast. How did you define the time –span for each single CTD acquisition? In the case of CTD 30 why 11 minutes and 29 sec?  The dynamic   of hydrothermal system can be very slow especially in case of diffuse vents. I am not sure that a time windows of few minutes is able to figure out in completely way the behaviour of a hydrothermal vents. Profile of CTD 30 highlights fluctuation only in short time range (high frequency), at the contrary for low frequency the profile does not show any fluctuation but a clear negative trend. It is very hard to define which is the regime of the volcano considering a short time series. I am very in doubt about your conclusion... "The absence of stationary behaviour (γ = 0) in temperature confirms that the effect of hydrothermal vent.

3)the mixing process: You described a possible effect of the oceanic seawater drifting in order to explain the intermittency in the conductivity. In addition the drifting would be responsible for variation of temperature. Considering the narrow area of investigation  and considering also that the extreme sampling  points, affected by conductivity intermittency behaviour are spaced few ten meters from others is it possible that mixing/drifting process can be responsible of fluctuation in temperature and Cond  over the entire area. My major doubt is that  dynamics along the seawater are very complicated. It is well know that seamount structure acts as obstacle for the seawater current triggering cyclonic upward flux of the current. Mixing/drifting process can represent a leading role in variation of T and C at the same way of a supposed hydrothermal vents.

I suggest to include in your study  a statistic analysis of a CTD cast acquired far from the seamount (i.e. at the base of the edifice) where the supposed hydrothermal exchanges is not affecting the seawater parameter. In this way you might exclude (or not)  the effect of seawater current mixing.

4)I have directly added some minor remarks on the pdf manuscript.

Reviewer 2 Report

See comments and suggestions in attached document.

Reviewer 3 Report

As I don't have enough knowledge about statistical processing to understand properly, I wasn't suitable for peer review of that part. In this paper, the observed data obtained are statistically processed to show that the substances released from the crater cause temperature and conductivity disturbances. Even 10 years after the eruption, it is shown that the anomaly is clearly present in the two variables, and the index for statistically evaluating them was also shown, and at the same time, the importance of continuous observation under the sea area on the active crater was shown. I think it is valuable for the progress of science and technology, but I don't think it shows any topical results. For the general readers (volcanologists) like the present reviewer unfamiliar to statistical processing, the methods and content of the discussion are hard to understand. I hope that it will be modified moderately so that it will be easier for general researchers to understand.

Round 2

Reviewer 1 Report

Dear Authors, many thanks for the revison work. All my remarks have been accounted. Now the manuscript is more rubust and  more readble. I think that it is suitable for publication

thank you